# Grammatical Error Correction via Mixed-Grained Weighted Training

Jiahao Li[1], Quan Wang[2*], Chiwei Zhu[1], Zhendong Mao[1], Yongdong Zhang[1]

[1]University of Science and Technology of China, Hefei, China
[2]MOE Key Laboratory of Trustworthy Distributed Computing and Service,
Beijing University of Posts and Telecommunications, Beijing, China
jiahao66@mail.ustc.edu.cn, wangquan@bupt.edu.cn
tanz@mail.ustc.edu.cn, zdmao@ustc.edu.cn, zhyd73@ustc.edu.cn

## Abstract

The task of Grammatical Error Correction (GEC) aims to automatically correct grammatical errors in natural texts. Almost all previous works treat annotated training data equally, but inherent discrepancies in data are neglected. In this paper, the inherent discrepancies are manifested in two aspects, namely, accuracy of data annotation and diversity of potential annotations. To this end, we propose MainGEC, which designs token-level and sentence-level training weights based on inherent discrepancies in accuracy and potential diversity of data annotation, respectively, and then conducts mixed-grained weighted training to improve the training effect for GEC. Empirical evaluation shows that whether in the Seq2Seq or Seq2Edit manner, MainGEC achieves consistent and significant performance improvements on two benchmark datasets, demonstrating the effectiveness and superiority of the mixed-grained weighted training. Further ablation experiments verify the effectiveness of designed weights of both granularities in MainGEC.

## 1 Introduction

The task of Grammatical Error Correction (GEC) aims to automatically correct grammatical errors in natural texts, which is extremely beneficial for language learners, such as children and non-native speakers (Bryant et al., 2022). The currently dominant neural GEC methods are categorized into two groups, *i.e.*, Seq2Seq methods and Seq2Edit methods. Seq2Seq methods treat GEC as a monolingual translation task, regarding errorful sentences as the source language and error-free sentences as the target language (Yuan and Briscoe, 2016; Sun et al., 2021; Zhang et al., 2022b). Seq2Edit methods treat GEC as a sequence tagging task, which predicts a tagging sequence of edit operations to perform correction (Awasthi et al., 2019; Omelianchuk et al., 2020; Tarnavskyi et al., 2022).

---

*Corresponding author: Quan Wang.

| Accuracy of Data Annotation | |
|---|---|
| Sample 1: | Their new house near the beach is very nice . |
| Annotation: | Your (✗) new house near the beach is very nice . |
| Sample 2: | I read your email yesterday but I had n't had the time to reply yet . |
| Annotation: | I read your email yesterday but I have (✓) n't had the time to reply until now (✗) . |
| Sample 3: | do you have best friend in your life ? |
| Annotation: | Do (✓) you have a best friend (✓) in your life ? |

| Diversity of Potential Annotations | |
|---|---|
| Sample 4: | Natural environment destroyed that is a people focus on frequently problem . |
| Annotation: | The natural environment is being destroyed . That is a problem people focus on frequently . |
| Alternative: | The natural environment is being destroyed , which is a problem people focus on frequently . |
| Sample 5: | Secondly there are not much variety of dessert mainly fruits and puddings . |
| Annotation: | Secondly , there is not a lot of variety in the desserts , mainly fruits and puddings . |
| Alternative: | Secondly , there are not many varieties of desserts , mainly fruits and puddings . |
| Sample 6: | One of my favourite books are Diary of a Wimpy Kid . |
| Annotation: | One of my favourite books is Diary of a Wimpy Kid . |

Table 1: Instances from the BEA-19 (Bryant et al., 2019) training set to show the discrepancies in the annotated training data. Erroneous annotations are in red, correct annotations are in blue, and multiple potential annotations are in green.

Whether in the Seq2Seq or Seq2Edit manner, almost all previous works treat annotated training data equally (Rothe et al., 2021; Tarnavskyi et al., 2022), that is, assigning the same training weight to each training sample and each token therein. However, inherent discrepancies in data are completely neglected, causing degradation of the training effect. Specifically, inherent discrepancies may be manifested in two aspects, *i.e.*, accuracy of data annotation and diversity of potential annotations. The discrepancy in **accuracy of data annotation** refers to the uneven annotation quality, which is

caused by differences in the annotation ability of annotators and the difficulty of samples (Zhang et al., 2022a). For example, in Table 1, Sample 1 and Sample 2 contain annotation errors to varying degrees, while the annotation of Sample 3 is completely correct. The discrepancy in **diversity of potential annotations** refers to the different amounts of potential reasonable alternatives to annotation. Usually, it differs due to different sentence structures or synonymous phrases. For example, Sample 4 and 5 potentially have multiple reasonable annotations, while Sample 6 probably only has a single reasonable annotation. Due to the above data discrepancies, training data should be distinguished during the training process, by being assigned well-designed weights.

In this paper, we propose **MainGEC** (*i.e.*, **M**ixed-gr**a**ined we**i**ghted trai**n**ing for **GEC**), which designs mixed-grained weights for training data based on inherent discrepancies therein to improve the training effect for GEC. First, we use a well-trained GEC model (called a teacher model) to quantify accuracy and potential diversity of data annotation. On the one hand, the accuracy of annotations is estimated by the generation probability of the teacher model for each target token, which represents the acceptance degree of the teacher model for the current annotation. Then, the quantified accuracy is converted into token-level training weights, as the accuracy of annotations may vary not only across samples but even across tokens in a single sample, *e.g.*, sample 2 in Table 1. On the other hand, the diversity of potential annotations is estimated by the information entropy of output distribution of the teacher model for each training sample, which actually represents the uncertainty, *i.e.*, diversity, of the target sentences that the teacher model is likely to generate. Then, the quantified potential diversity is converted into sentence-level training weights, considering that the potential annotations may involve the semantics and structures of the entire sentence. Finally, the token-level and sentence-level weigths constitute our mixed-grained weights for the training process.

Lichtarge et al. (2020) also considers to allocate training weights for samples. However, they only consider discrepancies in synthetic data and still treat human-annotated data equally, while the discrepancies we consider are across all data. Additionally, they only design sentence-level weighting, without token-level weighting considered in this paper. From another perspective, our method can be regarded as an "alternative" knowledge distillation method. Compared to Xia et al. (2022) applying general knowledge distillation on GEC, our method uses a teacher model to obtain mixed-grained training weights based on inherent discrepancies in data to guide the training process, rather than forcing the output distribution of the student model to be consistent with that of the teacher model.

We apply our mixed-grained weighted training to the mainstream Seq2Seq and Seq2Edit methods, and both of them achieve consistent and significant performance improvements on two benchmark datasets, verifying the superiority and generality of the method. In addition, we conduct ablation experiments, further verifying the effectiveness of the designed weights of both granularities. Besides, we conduct the comparative experiment with the general knowledge distillation method on GEC, verifying that our mixed-grained training weighting strategy outperforms the general knowledge distillation strategy.

The main contributions of this paper are summarized as follows: (1) We investigate two kinds of inherent discrepancies in data annotation of GEC for the first time, and propose MainGEC, which designs mixed-grained training weights based on the discrepancies above to improve the training effect. (2) The extensive empirical results show that MainGEC achieves consistent and significant performance improvements over the mainstream Seq2Seq and Seq2Edit methods on two benchmarks, proving the effectiveness and generality of our method for GEC.

## 2 Preliminary

This section presents the formulation of GEC task and currently mainstream Seq2Seq and Seq2Edit methods for GEC.

### 2.1 Problem Formulation

Grammatical Error Correction (GEC) is to correct grammatical errors in natural texts. Given an errorful sentence $X = \{x_1, x_2, \cdots, x_m\}$ with $m$ tokens, a GEC system takes $X$ as input, corrects grammatical errors therein, and outputs a corresponding error-free sentence $Y = \{y_1, y_2, \cdots, y_n\}$ with $n$ tokens. In general, the target sentence $Y$ often has substantial overlap with the source sentence $X$.

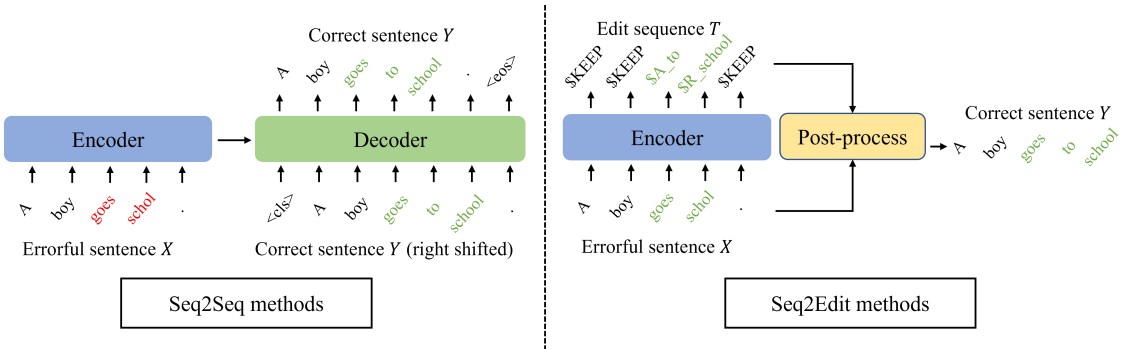

Figure 1: Overview of Seq2Seq and Seq2Edit methods for GEC. **Left:** Seq2Seq methods encode the errorful sentence $X$ by an encoder, and autoregressively generate the corresponding correct sentence $Y$ via a decoder. **Right:** Seq2Edit methods employ a sequence tagging model to predict a tagging sequence $T$ of edit operations corresponding to the errorful sentence $X$, and the correct sentence $Y$ is obtained by applying editing operations to $X$ via post-processing. Here, the tag *$A_to* denotes appending a new token "to" next to the current token "goes", and the tag *$R_school* denotes replacing the current token "schol" with "school".

## 2.2 Seq2Seq Methods

The Seq2Seq methods employ the encoder-decoder framework, where the encoder encodes the entire errorful sentence $X$ into corresponding hidden states, and the decoder autoregressively generates each token in $Y$ based on the hidden states and the previously generated tokens, as shown on the left in Figure 1.

The general objective function of the Seq2Seq methods is to minimize the negative log-likelihood loss:

$$\mathcal{L}(\theta) = -\sum_{i=1}^{n} \log p(\hat{y}_i = y_i | X, Y_{<i}, \theta),$$

where $\theta$ is learnable model parameters, $\hat{y}_i$ is the $i$-th token predicted by the model, and $Y_{<i} = \{y_1, y_2, \cdots, y_{i-1}\}$ denotes a set of tokens before the $i$-th token $y_i$.

## 2.3 Seq2Edit Methods

Due to the substantial overlap between $X$ and $Y$, autoregressive generation for the entire target $Y$ is inefficient, and Seq2Edit methods is a good alternative. The Seq2Edit methods usually employ a sequence tagging model made up of a BERT-like encoder stacked with a simple classifier on the top, as shown on the right in Figure 1. At first, a pre-defined set of tags is required to denote edit operations. In general, this set of tags contains universal edits, (*e.g. $KEEP* for keeping the current token unchanged, *$DELETE* for deleting the current token, *$VERB_FORM* for conversion of verb forms, *etc*)[1]

---

[1]Here we take GECToR's tags (Omelianchuk et al., 2020) for illustration.

and token-dependent edits, (*e.g. $APPEND_$e_i$* for appending a new token $e_i$ next to the current token, *$REPLACE_$e_i$* for replacing the current token with another token $e_i$). Considering the linear growth of tag vocab's size taken by token-dependent edits, usually, a moderate tag vocab's size is set to balance edit coverage and model size based on the frequency of edits. Then, the original sentence pair $(X, Y)$ is converted into a sentence-tags pair $(X, T)$ of equal length. Specifically, the target sentence $Y$ is aligned to the source sentence $X$ by minimizing the modified Levenshtein distance, and then converted to a tag sequence $T = \{t_1, t_2, \cdots, t_m\}$. Refer to Omelianchuk et al. (2020) for more details.

In training, the general objective function of the Seq2Edit methods is to minimize the negative log-likelihood loss for the tag sequence:

$$\mathcal{L}^{s2e}(\theta) = -\sum_{i=1}^{m} \log p(\hat{t}_i = t_i | X, \theta),$$

where $\hat{t}_i$ is the $i$-th tag predicted by the model. During inference, Seq2Edit methods predict a tagging sequence $\hat{T}$ at first, and then apply the edit operations in the source sentence $X$ via post-processing to obtain the predicted result $\hat{Y}$.

## 3 Our Approach

This section presents our approach, MainGEC which designs mixed-grained weights for training data based on inherent discrepancies therein to improve the training effect for GEC. Below, we first elaborate on how to quantify accuracy and potential diversity of data annotation at the token-level

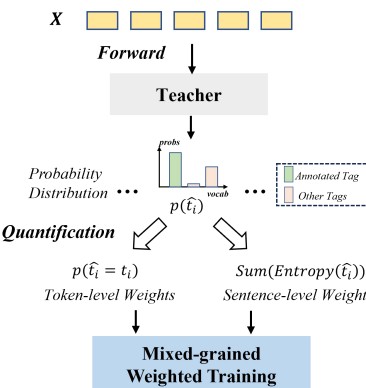

Figure 2: Illustration of MainGEC. MainGEC converts the target distribution generated by a teacher model and original targets into mixed-grained weights, and conducts weighted training with them.

and sentence-level respectively, and convert quantified features to training weights of both granularities, correspondingly. Then, based on both-grained weights, the overall mixed-grained weighted training strategy is introduced. Figure 2 summarizes the overall architecture of MainGEC.

## 3.1 Token-Level Weights

Due to differences in the annotation ability of annotators and the difficulty of samples, there is a discrepancy in the accuracy of data annotation. Actually, this discrepancy exists not only across samples but even across tokens in a single sample. To this end, a well-trained GEC model is used to quantify the accuracy of data annotation for each token in all training samples, and then they are converted into token-level training weights.

**For Seq2Seq Methods** The source sentence $X$ is fed into a well-trained Seq2Seq GEC model (called the teacher model), and the accuracy of the data annotation is estimated by the generation probability of the teacher model for each target token $y_i$ :

$$\boldsymbol{Acc}(y_i) = p(\hat{y}_i = y_i | X, Y_{<i}, \theta_T),$$

where $i \in \{1, 2, \cdots, n\}$, $\theta_T$ is parameters of the teacher model. Actually, this estimation implies the extend to which the teacher model agrees with the current annotation, which can be a measure of the accuracy. Then, quantified accuracy of data annotation for each target token can be directly regarded as the token-level training weight, as the higher accuracy of data annotation means the better annotation quality and thus a higher token-level training weight should be assigned for training. The

token-level training weights for Seq2Seq methods is defined as:

$$\boldsymbol{w}_{token}(y_i) = \boldsymbol{Acc}(y_i).$$

**For Seq2Edit Methods** Similarly, the accuracy of the data annotation is estimated by the generation probability of a well-trained Seq2Edit teacher model for each target tag $t_i$:

$$\boldsymbol{Acc}(t_i) = p(\hat{t}_i = t_i | X, \theta_T),$$

where $i \in \{1, 2, \cdots, m\}$. Correspondingly, the token-level training weights for each target tag is defined as:

$$\boldsymbol{w}_{token}(t_i) = \boldsymbol{Acc}(t_i).$$

## 3.2 Sentence-Level Weigths

Due to different sentence structures or synonymous phrases, there can be multiple potential reasonable alternatives to the single target sentence $Y$ of a training sample $(X, Y)$. Further, the amounts of potential reasonable alternatives may differ across all samples, which is referred to as the discrepancy in the diversity of potential annotations. Therefore, we quantify the diversity of potential annotations for each training sample by the same teacher model above, and convert them into sentence-level training weights.

**For Seq2Seq Methods** We feed the source sentence $X$ into the teacher model to obtain the probability distribution of its prediction result. For this sample $(X, Y)$, the diversity of potential annotations is estimated by the information entropy of this distribution:

$$\boldsymbol{Div}(X, Y) = \frac{1}{n} \sum_{i=1}^{n} \frac{\boldsymbol{H}(\hat{y}_i | X, Y_{<i}, \theta_T)}{log|V|},$$

where $|V|$ is the vocab size and $\boldsymbol{H}()$ denotes the entropy of a random variable, with $log|V|$ for normalization. Here, lower information entropy means that the teacher model produces a sparser and sharper probability distribution. This leads to the fact that fewer candidate target sentences are likely to be generated, *i.e.*, there is less diversity of potential annotations therein. Further, this means the teacher model has more confidence for the target annotation, and a higher sentence-level training weight should be assigned during training. Therefore, a monotonically decreasing function and proper boundary processing are applied

to the quantified diversity of potential annotations to obtain the sentence-level training weight for the sample $(X, Y)$:

$$\boldsymbol{w}_{sent}(X, Y) = \text{Max}[\frac{\log(\text{Div(X, Y)} + \epsilon)}{\log \epsilon}, \epsilon],$$

where $\epsilon$ is a small positive quantity (*e.g.*, $e^{-9}$).

**For Seq2Edit Methods**  Similarly, the diversity of potential annotations is estimated by the information entropy of output distribution of a Seq2Edit teacher model for a sample $(X, T)$:

$$\boldsymbol{Div}(X, T) = \frac{1}{m} \sum_{i=1}^{m} \frac{\boldsymbol{H}(\hat{t}_i | X, \theta_T)}{log|E|},$$

where $|E|$ is the size of the pre-defined tag set. Correspondingly, the sentence-level training weight for the sample $(X, T)$ is defined as:

$$\boldsymbol{w}_{sent}(X, T) = \text{Max}[\frac{\log(\text{Div(X, T)} + \epsilon)}{\log \epsilon}, \epsilon].$$

### 3.3   Mixed-Grained Weighted Training

The mixed-grained weighted training is to simply integrate both-grained weights into the training process. During training, the sentence-level weights determine the contribution of each sample to update the model parameters, while further token-level weights are used to adjust the importance of each token/tag therein.

**For Seq2Seq Methods**  We use the sentence-level and token-level weights as factors of the training loss for the samples and the tokens in them, respectively. The overall loss function of our mixed-grained weighted training is defined as:

$$\mathcal{L}_w(\theta) = - \sum_{(X,Y) \in D} \boldsymbol{w}_{sent}(X, Y) *$$
$$\sum_{i=1}^{n} \boldsymbol{w}_{token}(y_i) * \log p(\hat{y}_i = y_i | X, Y_{<i}, \theta),$$

where $D$ is all training corpus.

**For Seq2Edit Methods**  Similarly, the loss function of our MainGEC for Seq2Edit methods is defined as:

$$\mathcal{L}_w(\theta) = - \sum_{(X,T) \in D^T} \boldsymbol{w}_{sent}(X, T) *$$
$$\sum_{i=1}^{m} \boldsymbol{w}_{token}(t_i) * \log p(\hat{t}_i = t_i | X, \theta),$$

where $D^T$ is all training data after the tag transformation.

| Training Set | #Sent | #Tokens | #Errors |
|---|---|---|---|
| Troy-1BW | 1.2M | 30.9M | 100% |
| CLang-8 | 2.4M | 28.0M | 58% |
| NUCLE | 57K | 1.16M | 62% |
| FCE | 28K | 455K | 62% |
| W&I+LOCNESS | 34K | 628K | 67% |
| Test Set | #Sent | #Tokens | #Errors |
| CONLL-14 | 1.3K | 30.1K | 72% |
| BEA-19 | 4.5K | 85.7K | - |

Table 2: Statistics of the datasets, including the number of sentences, tokens, and the proportion of errorful sentences.

## 4   Experiments and Results

This section introduces our experiments and results on two benchmarks, *i.e.*, CONLL-14 (Ng et al., 2014) and BEA-19 (Bryant et al., 2019). Then, we conduct ablation experiments on both-grained training weights and comparative experiments with the general knowledge distillation method. Finally, a case study is presented to visualize the weights in MainGEC.

### 4.1   Experimental Setups

**Datasets and Evaluation Metrics**  As in Tarnavskyi et al. (2022), the training datasets we used consist of Troy-1BW (Tarnavskyi et al., 2022), CLang-8[2] (Rothe et al., 2021), NUCLE (Dahlmeier et al., 2013), FCE (Yannakoudakis et al., 2011), W&I+LOCNESS (Bryant et al., 2019). The statistics of the used datasets are shown in Table 2.

For evaluation, we consider two benchmarks, *i.e.*, CONLL-14 and BEA-19. CONLL-14 test set is evaluated by official $M^2$ scorer (Ng et al., 2014), while BEA-19 dev and test sets are evaluated by ERRANT (Bryant et al., 2017). Both evaluation metrics are precision, recall and $F_{0.5}$.

**Baseline Methods**  We compare MainGEC against the following baseline methods. All these methods represent current state-of-the-art on GEC, in a Seq2Seq or Seq2Edit manner.

**Seq2Seq Methods**

- *Lichtarge et al. (2020)* introduces a sentence-level training weighting strategy by scoring each sample based on delta-log perplexity, $\Delta ppl$, which represents the model's log per-

---

[2]Here, CLang-8 is a clean version of Lang-8 used in Tarnavskyi et al. (2022).

| Method | Model | Data Size | Architecture | CONLL-14 | | | BEA-19 | | |
|---|---|---|---|---|---|---|---|---|---|
| | | | | P | R | $F_{0.5}$ | P | R | $F_{0.5}$ |
| Seq2Seq | Lichtarge et al. (2020) | 340M | Transformer-big | 69.4 | 43.9 | 62.1 | 67.6 | 62.5 | 66.5 |
| | Stahlberg and Kumar (2021) | 540M | Transformer-big | 72.8 | 49.5 | 66.6 | 72.1 | 64.4 | 70.4 |
| | T5GEC (Rothe et al., 2021) | 2.4M | T5-large | - | - | 66.1 | - | - | 72.1 |
| | T5GEC (Rothe et al., 2021)[‡] | 2.4M | T5-xxl | - | - | 68.8 | - | - | 75.9 |
| | SAD (Sun et al., 2021) | 300M | BART (12+2) | - | - | 66.4 | - | - | 72.9 |
| | BART (Zhang et al., 2022b) | 2.4M | BART | 73.6 | 48.6 | 66.7 | 74 | 64.9 | 72.0 |
| | SynGEC (Zhang et al., 2022b) | 2.4M | BART + DepGCN | 74.7 | 49.0 | 67.6 | 75.1 | 65.5 | 72.9 |
| | BART (reimp)[†] | 2.4M | BART | 74.3 | **47.7** | 66.8 | 78.1 | 58.9 | 73.3 |
| | MainGEC (BART)[†] | | | **77.3** | 45.4 | **67.8** | **78.9** | **59.5** | **74.1** |
| Seq2Edit | PIE (Awasthi et al., 2019) | 1.2M | BERT-large | 66.1 | 43.0 | 59.7 | - | - | - |
| | GECToR (Omelianchuk et al., 2020) | 10.2M | XLNET-base | 77.5 | 40.1 | 65.3 | 79.2 | 53.9 | 72.4 |
| | TMTC (Lai et al., 2022) | 10.2M | XLNET-base | 77.9 | 41.8 | 66.4 | 81.3 | 51.6 | 72.9 |
| | GECToR-L (Tarnavskyi et al., 2022) | 3.6M | RoBERTa-large | 74.4 | 41.1 | 64.0 | 80.7 | 53.4 | 73.2 |
| | Lichtarge et al. (2020) (reimp) | 3.6M | RoBERTa-large | 76.4 | 40.5 | 64.9 | 80.4 | 54.4 | 73.4 |
| | GECToR-L (reimp) | 3.6M | RoBERTa-large | 75.9 | **40.2** | 64.4 | 80.9 | 53.3 | 73.3 |
| | MainGEC (GECToR-L) | | | **78.9** | 39.4 | **65.7** | **82.7** | **53.8** | **74.5** |

Table 3: Performance on the test sets of CONLL-14 and BEA-19, where precision (P), recall (R), $F_{0.5}$ ($F_{0.5}$) are reported (%). Baseline results are directly taken from their respective literatures. Results marked by "†" are obtained by applying a decoding approach (Sun and Wang, 2022) to adjust the precision-recall trade-off of inference, while the result marked by "‡" is not comparable here because it uses a much larger model capacity (11B parameters). **Note:** Better scores in MainGEC and the directly comparable baseline are bolded.

plexity difference between checkpoints for a single sample.

- *Stahlberg and Kumar (2021)* generates more training samples based on an error type tag in a back-translation manner for GEC pre-training.

- *T5GEC* (Rothe et al., 2021) pretrains large multi-lingual language models on GEC, and trains a Seq2Seq model on distillation data generated by the former more efficiently.

- *SAD* (Sun et al., 2021) employs an asymmetric Seq2Seq structure with a shallow decoder to accelerate training and inference efficiency of GEC.

- *BART* (Zhang et al., 2022b) applies a multi-stage fine-tuning strategy on pre-trained language model BART.

- *SynGEC* (Zhang et al., 2022b) extracts dependency syntactic information and incorporates it with output features of the origin encoder.

**Seq2Edit Methods**

- *PIE* (Awasthi et al., 2019) generates a tag sequence of edit operations and applys parallel decoding to accelerate inference.

- *GECToR* (Omelianchuk et al., 2020) defines a set of token-level transformations and conducts 3-stage training on a tagging model.

- *TMTC* (Lai et al., 2022) customizes the order of the training data based on error type, under GECToR's framework.

- *GECToR-L* (Tarnavskyi et al., 2022) applys Transfomer-based encoders of large configurations on GECToR.

**Implementation Details** For the Seq2Seq implementation, BART-large (Lewis et al., 2020) is choosed as the model backbone. At first, we fine-tune BART with vanilla training as the teacher model with fairseq[3] implementation. For a fair comparison with SynGEC (Zhang et al., 2022b), the training scheme here is to just fine-tune BART on the collection of all training sets excluding Troy-1BW dataset, for just one stage. More training details are discussed in Appendix A.

For the Seq2Edit implementation, we choose GECToR-L based on RoBERTa (Liu et al., 2019) as the model backbone. The checkpoint released by GECToR-L is used for the teacher model[4] to generate training weights of both granularities. We

---

[3] https://github.com/pytorch/fairseq
[4] Please refer to Appendix B for effect of different teachers on MainGEC.

| Model | CONLL-14 | | | BEA-19 | | |
|---|---|---|---|---|---|---|
| | P | R | $F_{0.5}$ | P | R | $F_{0.5}$ |
| BART | 74.3 | 47.7 | 66.8 | 78.1 | 58.9 | 73.3 |
| MainGEC | 77.3 | 45.4 | 67.8 | 78.9 | 59.5 | 74.1 |
| w/o Token | 74.3 | 48.0 | 67.0 | 79.0 | 57.6 | 73.6 |
| w/o Sent | 74.4 | 49.6 | 67.6 | 78.1 | 61.1 | 74.0 |
| GECToR-L | 75.9 | 40.2 | 64.4 | 80.9 | 53.3 | 73.3 |
| MainGEC | 78.9 | 39.4 | 65.7 | 82.7 | 53.8 | 74.5 |
| w/o Token | 74.4 | 43.1 | 64.9 | 81.2 | 53.1 | 73.4 |
| w/o Sent | 74.3 | 43.8 | 65.2 | 80 | 57.2 | 74.1 |

Table 4: Ablation results on MainGEC, with the Seq2Seq group at the top and the Seq2Edit group at the bottom. The following changes are applied to MainGEC: removing the token-level training weights (w/o Token), and removing the sentence-level training weights (w/o Sent).

| Method | CONLL-14 | | | BEA-19 | | |
|---|---|---|---|---|---|---|
| | P | R | $F_{0.5}$ | P | R | $F_{0.5}$ |
| GECToR-L | 75.9 | 40.2 | 64.4 | 80.9 | 53.3 | 73.3 |
| KD | 76.9 | **40.7** | 65.3 | 81.0 | **54.4** | 73.8 |
| MainGEC | **78.9** | 39.4 | **65.7** | **82.7** | 53.8 | **74.5** |

Table 5: Comparison between MainGEC and the general knowledge distillation method for GEC.

also conduct 3-stage training as in GECToR-L. In Stage I, the model is pretrained on the Troy-1BW dataset. Then, in Stage II, the model is fine-tuned on the collection of the CLang-8, NU-CLE, FCE, and W&I+LOCNESS datasets, filtered out edit-free sentences. In Stage III, the model is fine-tuned on the W&I+LOCNESS dataset. All training hyperparameters used in MainGEC are set to their default values as in GECToR-L. Besides, we re-implement the most closely-related work, Lichtarge et al. (2020), based on GECToR-L for a more equitable comparison.

All checkpoints are selected by the loss on BEA-19 (dev) and all experiments are conducted on 1 Tesla A800 with 80G memory.

## 4.2 Main Results

Table 3 presents the main results of Seq2Seq and Seq2Edit methods. We can see that whether in the Seq2Seq or Seq2Edit manner, MainGEC brings consistent performance improvements on both benchmarks, verifying the effectiveness of our method. Concretely, compared to vanilla training, our mixed-grained weighted training leads to 1.0/0.8 improvements in the Seq2Seq manner, and 1.3/1.2 improvements in the Seq2Edit manner. In addition, MainGEC outperforms all baselines on BEA-19 benchmark, with 1.2/1.3 improvements over previous SOTAs, while it also has a comparable performance on CONLL-14 benchmark. These results prove the superiority of our method.

## 4.3 Ablation Study

We also conduct ablation study on MainGEC to investigate the effects of both-grained training weights, in the Seq2Seq and Seq2Edit manners. Table 4 presents the ablation results. It is obviously observed that whether token-level or sentence-level training weights included in MainGEC, can bring a certain degree of improvement over the baseline. Moreover, the mixed-grained weighted training can provide more improvements on the basis of a single grained weighted training.

## 4.4 Exploration w.r.t Knowledge Distillation

As there is a "teacher" model used to obtain training weights in MainGEC, it is necessary to compare MainGEC with the general knowledge distillation method (Xia et al., 2022) for GEC, refered as KD. In KD, the probability distribution generated by the teacher model is regarded as a soft objective, which supervises the entire training process with the original groundtruth together. Here, we reimplement KD in the Seq2Edit manner, where the teacher model is the same as before and GECToR-L (RoBERTa-large) is choosed as the student model. The experimental result is presented in Table 5. As we can see, KD brings a significant improvement over the baseline, due to extra knowledge from the teacher model. More importantly, with the same teacher model, MainGEC outperforms KD with a considerable margin. This proves our our mixed-grained weighted training is superior to KD, forcing the output distribution of the student model to be consistent with that of the teacher model.

## 4.5 Case Study

Figure 3 shows the same cases as in Table 1 and their token-level or sentence-level weights obtained in MainGEC. The weights here are obtained in the Seq2Edit manner. As we can see, token-level and sentence-level weights in MainGEC indeed reflect the accuracy and potential diversity of data annotation respectively, to some extend. Specifically,

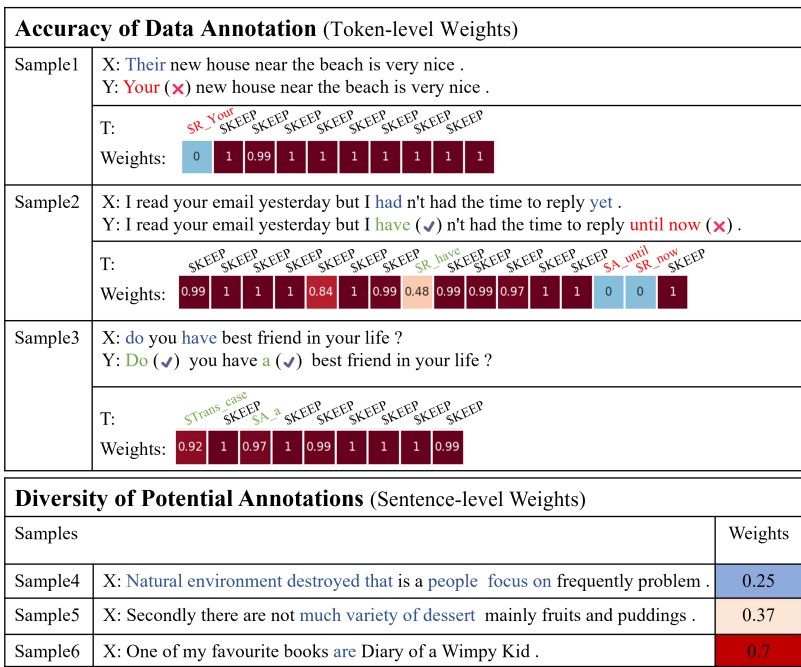

| Accuracy of Data Annotation (Token-level Weights) | | |
| --- | --- | --- |
| Sample1 | X: Their new house near the beach is very nice .
Y: Your (✗) new house near the beach is very nice .

T: $R\_Your$ $KEEP$ $KEEP$ $KEEP$ $KEEP$ $KEEP$ $KEEP$ $KEEP$ $KEEP$ $KEEP$
Weights: 0 \| 1 \| 0.99 \| 1 \| 1 \| 1 \| 1 \| 1 \| 1 \| 1 | |
| Sample2 | X: I read your email yesterday but I had n't had the time to reply yet .
Y: I read your email yesterday but I have (✓) n't had the time to reply until now (✗) .

T: $KEEP$ $KEEP$ $KEEP$ $KEEP$ $KEEP$ $KEEP$ $KEEP$ $R\_have$ $KEEP$ $KEEP$ $KEEP$ $KEEP$ $KEEP$ $A\_until$ $R\_now$ $KEEP$
Weights: 0.99 \| 1 \| 1 \| 1 \| 0.84 \| 1 \| 0.99 \| 0.48 \| 0.99 \| 0.99 \| 0.97 \| 1 \| 1 \| 0 \| 0 \| 1 | |
| Sample3 | X: do you have best friend in your life ?
Y: Do (✓) you have a (✓) best friend in your life ?

T: $Trans\_case$ $KEEP$ $A\_a$ $KEEP$ $KEEP$ $KEEP$ $KEEP$ $KEEP$ $KEEP$
Weights: 0.92 \| 1 \| 0.97 \| 1 \| 0.99 \| 1 \| 1 \| 1 \| 0.99 | |
| Diversity of Potential Annotations (Sentence-level Weights) | | |
| Samples | | Weights |
| Sample4 | X: Natural environment destroyed that is a people  focus on frequently problem . | 0.25 |
| Sample5 | X: Secondly there are not much variety of dessert  mainly fruits and puddings . | 0.37 |
| Sample6 | X: One of my favourite books are Diary of a Wimpy Kid . | 0.7 |

Figure 3: The samples in Table 1 and corresponding token-level or sentence-level weights obtained in MainGEC. For those token with problematic annotations or those samples with multiple potential appropriate annotations, MainGEC will assign relatively low token-level or sentence-level training weights, respectively. The correct annotations are in green, the erroneous annotations are in red, and the corresponding spans in the source sentences are in blue.

for those problematic annotation, MainGEC will assign a relatively low token-level weight, and vice versa. When there are multiple potential appropriate annotations for a single sample, only one objective contained in the training set will be assigned a relatively low sentence-level weight. For example, the sentence-level weights of Sample 4 and Sample 5 in Table 1 are relatively low due to multiple candidate sentence structures and synonymous phrases, respectively. This demonstrates that MainGEC is consistent with our motivation at first.

## 5   Related Works

GEC is a fundamental NLP task that has received wide attention over the past decades. Besides of the early statistical methods, the currently mainstream neural GEC methods are categorized into two groups, *i.e.*, Seq2Seq methods and Seq2Edit methods, in general.

Seq2Seq methods treat GEC as a monolingual translation task, regarding errorful sentences as the source language and error-free sentences as the target language (Yuan and Briscoe, 2016). Some works (Ge et al., 2018; Sun et al., 2022) generate considerable synthetic data based on the symmetry of the Seq2Seq's structure for data augmenta-

tion. In addition, some works (Kaneko et al., 2020; Zhang et al., 2022b) feed additional features into the neural network to improve GEC, such as the BERT (Devlin et al., 2019) presentation or syntactic structure of the input sentence.

Seq2Edit methods treat GEC as a sequence tagging task, which predicts a tagging sequence of edit operations to perform correction (Malmi et al., 2019). Parallel Iterative Edit (PIE) (Awasthi et al., 2019) and GECToR (Omelianchuk et al., 2020) define a set of tags representing the edit operations to be modelled by their system. Lai et al. (2022) investigates the characteristics of different types of errors in multi-turn correction based on GECToR. Tarnavskyi et al. (2022) applies multiple ensembling methods and knowledge distillation on the large version of the GECToR system.

## 6   Conclusion

This paper proposes MainGEC, which assigns mixed-grained weights to training data based on inherent discrepancies in data to improve the training effect for GEC. Our method uses a well-trained GEC model to quantify the accuracy and potential diversity of data annotation, and convert them into the mixed-grained weights for the training

process. Whether in the Seq2Seq or Seq2Edit manner, MainGEC achieves consistent and significant performance improvements on two benchmark datasets, verifying the superiority and generality of the method. In addition, further ablation experiments and comparative experiments with the general knowledge distillation method provide more insights on both-grained training weights and the perspective of knowledge distillation.

## Limitations

Our approach requires a well-trained model (called a teacher model) to obtain weights of two granularities before training. Therefore, compared to vanilla training, MainGEC has the additional preparation step to first acquire a teacher model (publicly released or trained by yourself) and then compute the weights by a forward propagation. In addition, the teacher model needs to be consistent with the weighted trained model in terms of type (Seq2Seq or Seq2Edit) and tokenizer.

## Acknowledgements

We would like to thank all the reviewers for their valuable advice to make this paper better. This research is supported by National Science Fund for Excellent Young Scholars under Grant 62222212 and the General Program of National Natural Science Foundation of China under Grant 62376033.

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

# A Training Details

The hyper-parameters for MainGEC (BART) are listed in Table 6.

| Configurations | Values |
|---|---|
| **Fine-tune** | |
| Model Architecture | BART-large |
| Number of epochs | 30 |
| Devices | 1 Tesla A800 with 80G |
| Max tokens per GPU | 20480 |
| Update Frequency | 2 |
| Learning rate | 3e-05 |
| Optimizer | Adam ($\beta_1 = 0.9, \beta_2 = 0.98, \epsilon = 1e-8$) |
| Learning rate scheduler | polynomial decay |
| Weight decay | 0.01 |
| Loss Function | cross entropy |
| Warmup | 2000 |
| Dropout | 0.3 |

Table 6: Hyper-parameters values for MainGEC (BART).

# B Effect of Different Teachers

In MainGEC, a teacher is used to quantify training weights of both granularities, which is the main contribution of this work. To investigate effect of different teacher on MainGEC, we conduct comparative experiments under two settings: (1) Teachers of different model scales: we use GECTOR (RoBERTa-base) and GECTOR (RoBERTa-large) as the teacher respectively for weighted training of GECTOR (RoBERTa-base). (2) MainGEC with self-paced learning: we use MainGEC as a stronger teacher for a new round of weighted training, i.e. iterative weighted training with MainGEC. The teacher used in the second round of training is the same model scale as the teacher used in the first round but performs better in GEC.

Table 7 and Table 8 present the experiment results respectively. Experiment results show that no matter what teacher model you use, mixed-grained weights generated by them can bring improvement over the baseline, verifying effectiveness of MainGEC. Besides, this improvement is not sensitive to the choice of the teacher, either with different model sizes or with different performances in GEC.

| Method | CONLL-14 | BEA-19 (dev) |
|---|---|---|
| GECToR (w/o teacher) | 63.4 | 52.9 |
| MainGEC (w/ base teacher) | 64.9 | **55.6** |
| MainGEC (w/ large teacher) | **65.1** | 54.9 |

Table 7: Performance of MainGEC based on teachers of different model scales.

| Method | CONLL-14 | BEA-19 (dev) |
|---|---|---|
| GECToR-L (original teacher) | 64.4 | 56.4 |
| MainGEC (1st round) | 65.7 | **57.6** |
| MainGEC (2nd round) | **66.1** | 57.4 |

Table 8: Performance of MainGEC with self-paced learning.