# OpenReview forum: "Grammatical Error Correction via Mixed-Grained Weighted Training"
_EMNLP/2023/Conference — EMNLP 2023 Findings_

### Official Review · Reviewer_uf6d · 2023-07-22

**Soundness:** 3

**Excitement:**

3: Ambivalent: It has merits (e.g., it reports state-of-the-art results, the idea is nice), but there are key weaknesses (e.g., it describes incremental work), and it can significantly benefit from another round of revision. However, I won't object to accepting it if my co-reviewers champion it.

**Paper Topic And Main Contributions:**

This paper introduces a mixed-grained weighted training approach for grammatical error correction (GEC). The authors identify inherent discrepancies in GEC training data, considering both annotation quality and diversity. They thereby propose weights at both the token and sample levels.

The empirical results demonstrate that this weighted training strategy enhances the performance of sequence-to-sequence (seq2seq) and sequence-to-edit (seq2edit) baselines, underscoring the efficacy of their proposed method.

**Questions For The Authors:**

None

**Reasons To Accept:**

1. the performance improvement over strong baselines seems good.
2. the proposed training method is easy to implement and apply.

**Reasons To Reject:**

1. The concept of weighted training, even when mixed-grained, is not particularly novel. This paper can be more accurately classified as an empirical contribution.
2. The comparison with Lichtarge et al. (2020) appears to be somewhat skewed. As this is the most closely-related work and serves as a significant baseline, it would be more appropriate to re-implement their method for a more equitable comparison. This would provide a stronger basis for asserting the superiority of your method.
3. To further validate the effectiveness of your method, it would be beneficial to conduct additional experiments in other languages, such as Chinese, as well as in low-resource languages like Czech. This would provide a more comprehensive understanding of the method's applicability and robustness.

**Reproducibility:**

4: Could mostly reproduce the results, but there may be some variation because of sample variance or minor variations in their interpretation of the protocol or method.

**Reviewer Confidence:**

4: Quite sure. I tried to check the important points carefully. It's unlikely, though conceivable, that I missed something that should affect my ratings.

---

> ### Author Rebuttal · Authors · 2023-08-28
>
> Thanks to the reviewer's valuable suggestion, we make some explanations here and will improve them in the latter version.
>
> ## Weakness #1
> Our contribution is to design a novel mixed-grained training weight based on discrepancies of GEC data to guide the training process under the GEC task. This has not been explored before.
>
> ## Weakness #2
> We have re-implemented the work of Lichtarge et al. (2020), and compared it with the baseline and MainGEC. The F_0.5 performance on conll14 and bea19 dev are as follows:
>
> | Methods | conll14 | bea19-dev |
> | :-----| :----: | :----: |
> | GECTOR-L | 64.4 | 56.4 |
> | Lichtarge et al. | 62.1 | - |
> | Lichtarge et al. (reimp) | 65.1 | 57.1 |
> | MainGEC | 65.7 | 57.6 |
>
> It can be seen that Lichtarget et al. (2020) also has a considerable performance improvement for the GEC model, while MainGEC improves more performance, which proves the superiority of our method.
>
> ## Weakness #3
> We leave additional experiments on other languages as future works.

---

### Official Review · Reviewer_45hm · 2023-08-05

**Soundness:** 3

**Excitement:**

3: Ambivalent: It has merits (e.g., it reports state-of-the-art results, the idea is nice), but there are key weaknesses (e.g., it describes incremental work), and it can significantly benefit from another round of revision. However, I won't object to accepting it if my co-reviewers champion it.

**Paper Topic And Main Contributions:**

This paper presents a teacher-based weighted training method for prevalent grammatical error correction (GEC) paradigms, including seq2seq and seq2edit.

The main contribution of this paper lies in its consideration of the intra-discrepancy of training data and the proposal of an effective solution to alleviate the potential negative effects caused by these discrepancies during the training stage. To distinguish different data during the training process, the authors suggest using a homologous teacher model to measure the accuracy (token level) and diversity (sentence level) of data annotation. The probabilistic scores generated by the teacher model are then used to assign weights to the training instances.

The results demonstrate that the proposed MainGEC method outperforms the baselines, and remains competitive when compared to the traditional teacher-student knowledge distillation method.


**Questions For The Authors:**

A. Do you think it is necessary to add further weight to balance the learning of two different granularities?

B. In the later training stage, the student model could evolve into a well-trained GEC model. Could we then utilize a well-trained student model to derive the weights? (This may be connected to other interesting topics, such as self-paced learning and curriculum learning.)


**Reasons To Accept:**

- **Novelty**: The proposed method introduces a novel approach by quantifying and addressing the discrepancy in training instances for the GEC task. This innovative perspective may inspire future research and provide valuable insights, especially considering the challenges posed by data scarcity in GEC system training.

- **Generality**: The MainGEC method exhibits a high level of generality and effectiveness, making it suitable for both seq2seq and seq2edit paradigms.

- **Clarity, Organization**: The paper's methodology is clearly presented, and the provided examples, such as Table 1 and Figure 3, aid in comprehending the problem and acknowledging the effectiveness of the proposed methods.


**Reasons To Reject:**

- **Flexibility**: One of the limitations of this approach lies in the requirement for a homologous teacher model in terms of paradigm and vocabulary table with the student model. This can hinder the method's flexibility and may pose challenges during the preparation of the teacher model.

- **Scope**: The pre-trained teacher model tends to be already a strong model; do we truly need to train a student model from scratch using weighted training? Is it possible to simply fine-tune the teacher model with weighted training? Weighted training would be meaningful if our goal is to obtain a small yet strong student model. However, the authors have not clearly indicated the specific targeted scenarios in this paper.

- **Experiments**:
  1) The authors did not explore how different teacher models affect the student's learning effectiveness, which could have provided valuable insights into the impact of varying teacher models on the proposed method's performance. The choice of teacher model may lack flexibility; however, it is worth noting that there are numerous robust GEC systems that share the same PLM architecture.
  2) The effectiveness of the proposed approach for other language families remains unknown.


**Reproducibility:**

4: Could mostly reproduce the results, but there may be some variation because of sample variance or minor variations in their interpretation of the protocol or method.

**Reviewer Confidence:**

4: Quite sure. I tried to check the important points carefully. It's unlikely, though conceivable, that I missed something that should affect my ratings.

**Typos Grammar Style And Presentation Improvements:**

- Line 183: REPLACE_ei -> *REPLACE_ei* (to be consistent with other expressions)
- Line 324: i.e. , -> i.e., (delete the redundant space)
- Line 387: Seq2Seq2Seq -> Seq2Seq

---

> ### Author Rebuttal · Authors · 2023-08-28
>
> We appreciate the constructive comments, and would like to make some clarifications which we hope could address the reviewer’s concerns. The preliminary experiments given here will be refined later and added to the revised paper.
>
> ## Weakness #1
> In fact, the preparation of the teacher model is not difficult. In general scenarios, you definitely need to train a usable GEC model. In our scenario, you can directly use this GEC model as a teacher model, generate mixed-grained weights, and then perform weighted training on an identical model that will perform better than the general trained GEC model, similar to the idea of self-distillation.
>
>
> ## Weakness #2
> The original intention of designing MainGEC is to improve the performance of the existing advanced GEC system, so we choose to train the model from scratch in order to get the strongest possible GEC model. Alternatively, our method can also be used to directly fine-tune the teacher model to improve performance.
>
> Experimentally, we only fine-tune GECTOR-L (teacher) with weighted training to get MainGEC (finetune). The performance comparison of F_0.5 on conll14 and bea19-dev is as follows:
>
> | Methods | conll14 | bea19-dev |
> | :-----| :----: | :----: |
> | GECTOR-L (teacher) | 64.4 | 56.4 |
> | MainGEC (finetune) | 65.6 | 56.4 |
> | MainGEC (from scratch) | 65.7 | 57.6 |
>
> It can be seen that MainGEC (finetune) also has a certain performance improvement compared to the teacher model. Besides, it sacrifices part of the performance improvement compared to the MainGEC trained from scratch, with saving some training cost.
>
> Therefore, if you are pursuing training efficiency, you can also choose to use the MainGEC method to fine-tune the teacher only. Even so, it can also bring about a certain performance improvement, while more improvement is brought if you train a student model with MainGEC from scratch.
>
> ## Weakness #3
> We conduct preliminary supplementary experiments to explore the effect of different teacher models, using GECTOR (RoBERTa -large) and GECTOR (RoBERTa -base) as the teacher model for weighted training of GECTOR (RoBERTa -base). The performance comparison of F_0.5 on conll14 and bea19-dev is as follows:
>
> | Methods | conll14 | bea19-dev |
> | :-----| :----: | :----: |
> | GECTOR (w/o teacher) |63.4 | 52.9 |
> | MainGEC (w base teacher) | 64.9 | 55.6 |
> | MainGEC (w large teacher) | 65.1 | 54.9 |
>
> It can be seen that the mixed-grained weights generated by different teacher models are both improved for students, and this improvement is not sensitive to the size of the teacher model itself.
> We leave additional experiments on other languages as future works.
>
>
> ## Question #1
> We think that there may be no need to design a further weight. In our method, the impacts of both-grained weights on the training objective appear to be complementary. Specifically, the importance of each sample to model optimization is measured by the sentence-level weight. And, each token in a sample shares the sentence-level weight while reflecting its own importance by the token-level weight. Experimental results also verify its effectiveness, without a further weight.
>
> In addition, there are some difficulties in introducing a further weight. Formally, these two weights have a product relationship in the objective function. As far as we know, there is no general weighting method for this case. And, designing a further weight itself will introduce additional complexity, including specific selection of weighting methods and weighting value.
>
> ## Question #2
> In fact, we conducted an experiment in which the stronger MainGEC model was used as a new teacher model to guide a new round of weighted training process, that is, self-paced learning. The experimental results are as follows:
>
> | Methods | conll14 | bea19-dev |
> | :-----| :----: | :----: |
> | GECTOR-L (original teacher) |64.4 | 56.4 |
> | MainGEC (w/ origin teacher) | 65.7 | 67.6 |
> | MainGEC (w/ stronger teacher) | 66.1 | 57.4 |
>
> The results show that the newly stronger teacher does not bring significant further performance improvement to MainGEC. This is consistent with the above experimental conclusion "MainGEC is not sensitive to the selection of the teacher".

---

### Official Review · Reviewer_6raQ · 2023-08-08

**Soundness:** 3

**Excitement:**

3: Ambivalent: It has merits (e.g., it reports state-of-the-art results, the idea is nice), but there are key weaknesses (e.g., it describes incremental work), and it can significantly benefit from another round of revision. However, I won't object to accepting it if my co-reviewers champion it.

**Paper Topic And Main Contributions:**

The paper proposes MainGEC, a method for modeling (1) annotation accuracy and (2) potential diversity in GEC data, using token-level and sentence-level training weights. The training weights are computed by first pass trained GEC model: generation probability for each target token (token level weight) is used to estimate (1), entropy of the distribution of output sentences (sentence level weight) is used to model (2).

The paper reports positive results from this approach on two GEC benchmarks, with seq-to-seq and tagging models.


**Questions For The Authors:**

Question A: Why do you think this approach is particularly suited for GEC compared to other text generation tasks?

**Reasons To Accept:**

* The data weighting methods proposed by the authors gives some improvements over the baselines they selected.
* The paper presents useful ablations and an interesting comparison to knowledge distillation.
* The concrete example in Figure 3 is also quite nice to see.


**Reasons To Reject:**

* The method described by the authors could be equally applied to any seq-to-seq task. The paper would be stronger if it showed similar improvements in other language tasks, maybe machine translation, data-to-text, or other text generation tasks. If the weighting mechanism proposed only works for GEC, it could be a good research question to find out why.
* The paper fails to report SotA T5GEC-XXL results in their main results table (Table 3), which are much better than theirs. Lines 345-348 also claim that the methods listed in the paper represent the current SotA on GEC, which is not true. The paper would be much stronger if it showed improvements at the scale of T5GEC-XXL, or at least the paper should note that this model was outside their computational budget.


**Reproducibility:**

5: Could easily reproduce the results.

**Reviewer Confidence:**

5: Positive that my evaluation is correct. I read the paper very carefully and I am very familiar with related work.

**Typos Grammar Style And Presentation Improvements:**

* It seems that “discrepancies” might not be the right word to use here. Line 48-50 claims that accuracy and diversity are the only discrepancies, but there are many discrepancies in GEC data, much more than just accuracy of annotations and multiple potential annotations. What about domain or topic differences? Or speaker first language and speaker proficiency? Or access to a GEC model while writing?
* The examples on the first page are not very convincing. The alleged annotation error in Sample 1 could be a justified correction depending on context. The alleged annotation error Sample 2 could also be justified because it makes the sentence more clear. Sample 6 also does not seem to showcase multiple potential annotations, so it should not be in green according to the caption.

---

> ### Author Rebuttal · Authors · 2023-08-28
>
> We would like to thank the reviewer for the constructive comments and answer the questions as follows.
>
> ## Weakness #1 (Question #1)
> Our method is originally designed for the GEC task. If it is considered to be applied to other text generation tasks, the performance improvement may be not as significant as that of GEC. The reason is explained as follows:
>
> For data annotation in GEC, the distribution of each target token is sparse and sharp, only on one or a few tokens, due to the limitation of the principle of minimal editing. Then, it is relatively easy for the teacher model to accurately fit these distributions, and thus appropriate weights can be given to guide training. Conversely, in other generative tasks (such as machine translation), the probability of each target token is distributed over a wider range of tokens due to the freedom of text expression and word usage. Therefore, it is difficult to accurately fit the probability distribution of each token to obtain training weights, which finally affects the performance gains from weighted training.
>
> Therefore, if MainGEC is applied to other generation tasks, its performance improvement may not be as significant as that of GEC.
>
>
> ## Weakness #2
> It does exceed our computational overhead to conducting experiments with full data on the 11B T5-xxl. Therefore, our experiments are set on the model scale commonly used in existing advanced methods on GEC (BART: 400M, RoBERTa-large: 340M). For T5GEC, we choose T5GEC-large of the same magnitude but still about twice as large (770M) as a performance comparison. From the experimental results, it can be seen that MainGEC can still achieve relatively higher performance with a smaller number of model parameters.
>
> We will modify the conclusion of SOTA to best performance under model sizes of the same order in subsequent revisions.

---

### Meta-Review · Area_Chair_62W4 · 2023-09-20

**Recommendation:** 3

**Metareview:**

The paper addresses two issues with the training data used in Grammatical Error Correction (GEC) – namely, errors in annotation and the lack of annotation diversity. The paper proposes an approach that uses a weighting schema both on the token and sentence level to address these issues. Experiments on English benchmarks with the frameworks of seq2seq and seq2edit modeling paradigms demonstrate improvements over the baseline models.

The proposed method is interesting and the experimental results demonstrate the utility of the approach. The paper can be significantly strengthened by presenting experimental results on other languages. The paper should also make it clear that the obtained results are not SOTA. If larger models (such as T5-xxl) cannot be used due to their computational cost, the paper should nonetheless include those results (along with the SOTA results on the benchmarks).

---

### Decision · Program_Chairs · 2023-10-07

**Decision:**

Accept-Findings

**Comment:**

The paper addresses two issues with the training data used in Grammatical Error Correction (GEC) – namely, errors in annotation and the lack of annotation diversity. The paper proposes an approach that uses a weighting schema both on the token and sentence level to address these issues. Experiments on English benchmarks with the frameworks of seq2seq and seq2edit modeling paradigms demonstrate improvements over the baseline models.

The proposed method is interesting and the experimental results demonstrate the utility of the approach. The paper can be significantly strengthened by presenting experimental results on other languages. The paper should also make it clear that the obtained results are not SOTA. If larger models (such as T5-xxl) cannot be used due to their computational cost, the paper should nonetheless include those results (along with the SOTA results on the benchmarks).